

# Factors associated with the intention to vaccinate and price sensitivity to the human papillomavirus (HPV) vaccine among Chinese male medical college students: a cross-sectional survey

Yuan Li[1], Hiromi Kawasaki[1], Zhengai Cui[2,3] and Sae Nakaoka[1]

[1] Division of Integrated Health Sciences, Graduate School of Biomedical and Health Sciences, Hiroshima University, Hiroshima, Japan
[2] Shunde Women and Children's Hospital of Guangdong Medical University, Foshan, China
[3] Department of Management and Law, School of Humanities and Management, Guangdong Medical University, Dongguan, China

## ABSTRACT

**Background**. College students, particularly males, face a high risk of the human papillomavirus (HPV) infection, yet vaccination rates remain low in China. This study aims to explore the willingness to receive HPV vaccine among Chinese male medical college students based on health literacy (HL) theory and price sensitivity.

**Methods**. A cross-sectional survey was conducted from June 16 to July 16, 2024, to assess socioeconomic status and HL among college students at Guangdong Medical University. The survey was distributed via WeChat using convenience sampling through the Chinese online survey platform "Wenjuanxing" (http://www.wjx.cn). A Van Westendorp price sensitivity meter (PSM) was used to determine male medical college students' price sensitivity to HPV vaccines.

**Results**. Among 1,631 valid participants, 59.4% ($n = 969$) expressed willingness to receive HPV vaccination. The key influencing factors included graduate status (OR: 1.82; 95% CI [1.00–3.32]; $p = 0.049$), family history of cancer (OR: 1.29; 95% CI [1.01–1.66]; $p = 0.041$), moderate daily exercise (OR: 53; 95% CI [1.21–192]; $p < 0.001$), peers' HPV vaccination status (OR: 1.42; 95% CI [1.15–1.75]; $p < 0.001$), and HL levels (OR: 1.03; 95% CI [1.01–1.05]; $p = 0.002$). The Van Westendorp PSM analysis demonstrated high price sensitivity: the market price of the quadrivalent vaccine aligned closely with the lowest price point for male medical college students, whereas the nine-valent vaccine's market price exceeded the acceptable price range.

**Conclusions**. HL levels directly increased HPV vaccination intentions, and PSM analysis revealed the nine-valent vaccine's market price exceeded the acceptable price range, contrasting with the affordability-aligned quadrivalent vaccine. Interventions should prioritize HL programs and cost-reduction strategies (*e.g.*, subsidies for the nine-valent vaccine), while addressing nonprice barriers (including family history of cancer, moderate daily exercise and peer influence networks), particularly for the quadrivalent vaccine.

Corresponding author
Hiromi Kawasaki,
khiromi@hiroshima-u.ac.jp

## INTRODUCTION

Human papillomavirus (HPV) is the most prevalent sexually transmitted infection worldwide, and causes cervical, anal, vulvar, vaginal, penis, and head and neck cancers (*Bruni et al., 2023*). While HPV vaccination effectively prevents HPV-related diseases in both sexes, current HPV vaccination strategies predominantly target females, creating a significant prevention gap (*Milano et al., 2023*). Notably, males are often overlooked in vaccination programs despite their critical role in transmission dynamics, particularly in China where male HPV vaccination remains virtually nonexistent despite high infection rates (*Mao et al., 2025*; *Zhang et al., 2025*).

In China, the prevalence of HPV in males is 52.45% among outpatients and 7.89% in health checkups, indicating a high prevalence of infection. However, vaccination coverage in this population remains negligible (*Li et al., 2025*). Male college students, as sexually active and high-risk individuals, are vulnerable to HPV infection and its consequences (*Fu et al., 2022*; *WHO, 2022*). Vaccinating males not only reduces their risk of cancers such as anal and penile cancers but also protects female partners by curbing HPV transmission, thereby assisting in the prevention of cervical cancer (*Bergman et al., 2022*; *Rosado et al., 2023*). These dual benefits underscore why male vaccination should be prioritized, yet implementation barriers remain poorly understood.

However, the current literature reveals three critical knowledge gaps: First, data on HPV vaccination among male college students in China remain scarce (*Dai et al., 2023*). Second, existing studies predominantly focus on general populations while systematically neglecting the unique behavioral determinants of medical students who possess professional health knowledge but face practice-behavior gaps. Third, no studies have comprehensively examined how China's non-subsidized vaccination context and associated economic constraints affect male vaccination decisions—a gap our research directly tackles through novel price sensitivity analysis.

Studies consistently link higher health literacy (HL) with increased engagement in health-promoting behaviors, particularly HPV vaccination, which is a key preventive measure (*Jordan et al., 2025*; *Weiss, Doak & Doak, 2003*). However, few domestic studies have explored the association between HL and HPV vaccination intentions among male medical college students (*Dai et al., 2023*; *Lorini et al., 2018*; *Wu et al., 2023*).

To date, China has not fully integrated HPV vaccination into its national immunization program. As of January 8, 2025, only one type of HPV vaccine, the quadrivalent vaccine (imported), had been approved for males aged 9–26 years in mainland China, with a price of 831 CNY (equivalent to \$116.71 based on the exchange rate of 1 USD = 7.12 CNY as of June 16, 2024; the same exchange rate applies hereafter)—a substantial financial burden for most college students. We hypothesize that: (1) HL would be a key predictor of HPV

vaccination intentions among male medical students in China, and (2) price sensitivity would be an important reference for vaccination decisions.

This necessitates examining psychosocial determinants (*e.g.*, HL) and economic barriers (*e.g.*, pricing) within China's healthcare context. Originally developed for market research, the Van Westendorp price sensitivity meter (PSM) has become an established methodology for evaluating consumers' perceptions of pricing (*Van Westendorp, 1976*). This approach, which has been recognized for its simplicity and effectiveness in determining optimal price points for various goods and services (*Larson et al., 2014*; *Murtiningrum, Darmawan & Wong, 2022*), offers valuable potential for investigating price sensitivity related to HPV vaccination among male college students. To our knowledge, this is the first application of PSM in HPV vaccine research targeting Chinese males, bridging methodological innovations between marketing science and public health.

To test these hypotheses, this study specifically examines male medical college students in China—a population that presents a unique paradox of high health knowledge but potentially low vaccination uptake and that faces unique barriers to vaccination despite their elevated risk. This study aims to investigate vaccination intentions and price sensitivity to the HPV vaccine among male medical college students in Dongguan, China, specifically addressing two unresolved questions: (1) How does the HL component influence HPV vaccination intentions among male medical students? (2) What is the acceptable price range (APR) that balances vaccine affordability and perceived value in China's unique socioeconomic context?

## MATERIALS & METHODS

### Survey design and study participants

This study was conducted at Guangdong Medical University from June 16 to July 16, 2024. Using a convenience sampling method, an online questionnaire survey was administered through the Chinese online survey platform "Wenjuanxing" (http://www.wjx.cn). The survey was distributed to full-time students at the university *via* WeChat. The inclusion criteria were as follows: (1) full-time enrolled students; (2) completion and successful submission of all mandatory questionnaire items, excluding questions skipped due to branching logic. The exclusion criteria were as follows: (1) a response time of less than 180 s; (2) inconsistencies and illogical responses in some answers; and (3) failure to provide accurate academic and professional information.

A total of 5,384 individuals participated in the survey, with 456 responses excluded due to unmet criteria. This yielded 4,928 valid questionnaires (validity rate: 91.53%). Among the valid responses, 3,297 participants were female college students and 1,631 were male college students. As the analysis specifically targeted Chinese male college students, the final analysis comprised 1,631 participants.

### Ethical considerations

This study was approved by the Clinical Research Ethics Committee of the Affiliated Hospital of Guangdong Medical University (ethical application ref: KT2023-126-01, approval number: PJKT2023-126). The study participants read and signed an online

informed consent form. Potential study participants were informed that participation in this study was voluntary and that they could withdraw their decision to participate at any time.

## Variables and measurements

The survey included (1) socioeconomic and demographic characteristics, (2) the eHealth literacy scale, and (3) the Van Westendorp PSM.

## Socioeconomic and demographic characteristics

The following characteristics were assessed in this survey (*Dapari et al., 2024*; *Deng, Chen & Liu, 2021*; *Doğan et al., 2024*): age, education level, household registration, major, family history of cancer, sex education, smoking, exercise per day, alcohol consumption, HPV vaccination status of peers, and willingness to receive the HPV vaccine (Table 1).

## eHealth literacy scale

The Chinese adaptation of the 8-item eHealth literacy (eHL) scale originally validated by Dong and colleagues served as our measurement tool (*Dong et al., 2023*) (Table 2). The authors obtained permission to use this instrument from the copyright holders. A 5-point Likert-type scoring system was used (1 = strongly disagree to 5 = strongly agree), and cumulative scores were interpreted as indicating progressively higher eHL competencies. Internal consistency analysis revealed excellent scale reliability (Cronbach's $\alpha = 0.949$).

## Van Westendorp PSM

The Van Westendorp PSM test comprised four sequential phases. First, participants who were willing to receive the HPV vaccine selected their preferred type of vaccine (quadrivalent vaccine (imported), nine-valent vaccine (imported), or alternatives) based on provided product information (*Chen et al., 2024*; *Zhao et al., 2024*).

Second, the participants were required to provide four key pricing judgments regarding the selected vaccines (*Larson et al., 2014*), namely (1) a price so low it raises quality concerns (too cheap); (2) a price that is considered fair and reasonable (cheap); (3) a price that feels high but is still acceptable (expensive); and (4) a price deemed unaffordable (too expensive). Systematic analysis of responses to these four key pricing judgments enables the researchers to identify frequencies of product perception across price points and compute their cumulative percentages.

Third, we employed the Van Westendorp PSM function of "Wenjuanxing" to calculate key pricing metrics, including the highest pricing point (HPP), acceptable price point (APP), optimal price point (OPP), and lowest pricing point (LPP), along with APRs for the quadrivalent vaccine (imported) and nine-valent vaccine (imported) (*Murtiningrum, Darmawan & Wong, 2022*).

Finally, we evaluated the difference between OPP and APP for each vaccine as a measure of price sensitivity.

## Data analysis methods

Data analysis was performed using SPSS 29.0 (IBM Corp.). Continuous variables were summarized as the mean (M) $\pm$ standard deviation (SD), whereas categorical variables

**Table 1 Socioeconomic and demographic characteristics of the participants.**

| Variable | *n* (%) |
|---|---|
| Age (M ± SD) | 20.89 ± 1.89 |
| Education level | |
|     Undergraduate | 1,570 (96.3) |
|     Postgraduate | 61 (3.7) |
| Household registration | |
|     Rural | 940 (57.6) |
|     Cities and Towns | 691 (42.4) |
| Major | |
|     Medical-related majors | 1,214 (74.4) |
|     Nonmedical majors | 417 (25.6) |
| Family history of cancer | |
|     No | 1,244 (76.3) |
|     Yes | 387 (23.7) |
| Have you received sex education? | |
|     No | 265 (16.2) |
|     Yes | 1,366 (83.8) |
| Do you smoke? | |
|     No | 1,532 (93.9) |
|     Yes | 99 (6.1) |
| Exercise per day | |
|     <1 h | 1,084 (66.5) |
|     1–3 h | 503 (30.8) |
|     >3 h | 44 (2.7) |
| Alcohol consumption | |
|     Never | 636 (39.0) |
|     Occasionally | 943 (57.8) |
|     Often (more than once a week) | 52 (3.2) |
| Have any of your family, friends, or classmates gotten or reserved the HPV vaccine? | |
|     No | 713 (43.7) |
|     Yes | 832 (51.0) |
| Willingness to receive HPV vaccination if possible? | |
|     No | 663 (40.6) |
|     Yes | 968 (59.4) |

Notes.
$N = 1,631$.
% = n/N.

were reported as counts (*n*) and proportions (%). Univariate analyses were used to examine the effects of sociodemographic factors and HL on the willingness to receive HPV vaccination. Predictive determinants were identified *via* binary logistic regression, with effect sizes expressed as adjusted odds ratios (ORs) and 95% confidence intervals (CIs). Statistical significance was set at $p < 0.05$.
**Table 2  Descriptive statistics of eHealth literacy.**

| Term | Strongly disagree n(%) | Disagree n(%) | Neutral/Unsure n(%) | Agree n(%) | Strongly agree n(%) | Score (M ± SD) |
|---|---|---|---|---|---|---|
| 1. I know what health resources are available on the internet. | 28 (1.7) | 38 (2.3) | 263 (16.1) | 933 (57.2) | 369 (22.6) | 3.97 ± 0.80 |
| 2. I know where to find helpful health resources on the internet. | 29 (1.8) | 59 (3.6) | 308 (18.9) | 890 (54.6) | 345 (21.2) | 3.90 ± 0.83 |
| 3. I know how to find helpful health resources on the internet. | 33 (2.0) | 51 (3.1) | 288 (17.7) | 910 (55.8) | 349 (21.4) | 3.91 ± 0.83 |
| 4. I know how to use the internet to answer my health questions. | 33 (2.0) | 54 (3.3) | 278 (17.0) | 924 (56.7) | 342 (21.0) | 3.91 ± 0.83 |
| 5. I know how to use the health information I find on the internet to help me. | 29 (1.8) | 29 (1.8) | 206 (12.6) | 999 (61.3) | 368 (22.6) | 4.01 ± 0.76 |
| 6. I have the skills I need to evaluate the health resources I find on the internet. | 32 (2.0) | 54 (3.3) | 331 (20.3) | 865 (53.0) | 349 (21.4) | 3.89 ± 0.85 |
| 7. I can tell high-quality from low-quality health resources on the internet. | 31 (1.9) | 47 (2.9) | 305 (18.7) | 873 (53.5) | 375 (23.0) | 3.93 ± 0.84 |
| 8. I feel confident in using information from the internet to make health decisions. | 30 (1.8) | 49 (3.0) | 274 (16.8) | 905 (55.5) | 373 (22.9) | 3.95 ± 0.83 |
| eHealth Literacy Scores | | | | | | 31.46 ± 5.63 |

**Notes.**
$N = 1{,}631$.
$\% = n/N$.

# RESULTS

## Socioeconomic and demographic characteristics of the participants

A total of 1,631 male college students were included as valid participants in this study, with a mean age of $20.89 \pm 1.89$ years. Of the participants, 96.3% (1,570/1,631) were undergraduates, whereas 3.7% (61/1,631) were graduate students. Additionally, 57.6% (940/1,631) were from rural areas, and 74.4% (1,214/1,631) were medical students.

Regarding health-related characteristics, 23.7% (387/1,631) reported a family history of cancer, and 83.8% (1,366/1,631) had received sex education. The prevalence of smoking was 6.1% (99/1,631), and 57.8% (943/1,631) reported occasional alcohol consumption. In terms of physical activity, 66.5% (1,084/1,631) exercised less than one hour per day. Among participants who were aware of HPV, 51% (832/1,631) knew someone in their social circle who had either received or scheduled an HPV vaccination.

The findings revealed that 59.4% (968/1,631) of the male college students were willing to receive the HPV vaccine, whereas 40.6% (663/1,631) were unwilling (Table 1).

## eHealth literacy

The male college students in the sample achieved a mean total eHL measurement of $31.46 \pm 5.63$. This total corresponds to a per-item average of 3.93, which aligns with established thresholds for advanced eHL (Table 2).

## Univariate analyses

Univariate analysis revealed significant associations between willingness to receive HPV vaccination and key variables, as detailed in Table 3. Education level ($p = 0.038$), family history of cancer ($p = 0.022$), sex education ($p = 0.026$), exercise per day ($p < 0.001$), and the HPV vaccination status of peers ($p < 0.001$) demonstrated particularly significant correlations. Notably, the eHL level was significantly associated with vaccination willingness ($p < 0.001$).

**Table 3** Univariate analysis of willingness to receive HPV vaccination for the HPV vaccine among male college students.

| Variable | No (N = 663) | Yes (N = 968) | p value |
|---|---|---|---|
| **Socioeconomic and Demographic Characteristics** | | | |
| Age (M ± SD) | 20.91 ± 1.81 | 20.87 ± 1.94 | 0.417[b] |
| Education level | | | **0.038[a]** |
|     Undergraduate | 646 (97.4) | 924 (95.5) | |
|     Postgraduate | 17 (2.6) | 44 (4.5) | |
| Household Registration | | | 0.910[a] |
|     Rural | 381 (57.5) | 559 (57.7) | |
|     Cities and Towns | 282 (42.5) | 409 (42.3) | |
| Major | | | 0.272[a] |
|     Medical-related majors | 503 (75.9) | 711 (73.5) | |
|     Nonmedical majors | 160 (24.1) | 257 (26.5) | |
| Family history of cancer | | | **0.022[a]** |
|     No | 525 (79.2) | 719 (74.3) | |
|     Yes | 138 (20.8) | 249 (25.7) | |
| Have you received sex education? | | | **0.026[a]** |
|     No | 124 (18.7) | 141 (14.6) | |
|     Yes | 539 (81.3) | 827 (85.4) | |
| Do you smoke? | | | 0.561[a] |
|     No | 620 (93.5) | 912 (94.2) | |
|     Yes | 43 (6.5) | 56 (5.8) | |
| Exercise per day | | | **<0.001[b]** |
|     <1 h | 480 (72.4) | 604 (62.4) | |
|     1–3 h | 167 (25.2) | 336 (34.7) | |
|     >3 h | 16 (2.4) | 28 (2.9) | |
| Alcohol consumption | | | 0.234[b] |
|     Never | 270 (40.7) | 366 (37.8) | |
|     Occasionally | 373 (56.3) | 570 (58.9) | |
|     Often (more than once a week) | 20 (3.0) | 32 (3.3) | |
| Have any of your family, friends, or classmates gotten or reserved the HPV vaccine? | | | **<0.001[a]** |
|     No | 325 (52.6) | 388 (41.9) | |
|     Yes | 293 (47.4) | 539 (58.1) | |
| **eHealth Literacy Scores (M ± SD)** | 30.71 ± 5.65 | 31.98 ± 5.57 | **<0.001[b]** |

Notes.
[a] p value from the chi-square test.
[b] p value from the Mann–Whitney U test.
Bold indicates statistical significance ($p < 0.05$).

## Factors associated with willingness to receive HPV vaccination

The variables that were found to be significant in the univariate analysis were subsequently included in a binary logistic regression analysis. The results are presented in Table 4.

The following socioeconomic and demographic factors were identified as predictors of the willingness to receive HPV vaccination: education level, family history of cancer,

**Table 4   Binary logistic regression for willingness to receive HPV vaccination among male college students.**

| Variable | Logit coefficient | OR (95% CI) | $p$ value |
|---|---|---|---|
| **Socioeconomic and Demographic Characteristics** | | | |
| Education level | | | |
|     Undergraduate | Ref | | |
|     Postgraduate | 0.60 | 1.82 (1.00–3.32) | **0.049** |
| Family history of cancer | | | |
|     No | Ref | | |
|     Yes | 0.26 | 1.29 (1.01–1.66) | **0.041** |
| Have you received sex education? | | | |
|     No | Ref | | |
|     Yes | 0.24 | 1.27 (0.95–1.69) | 0.106 |
| Exercise per day | | | **0.002** |
|     <1 h | Ref | | |
|     1–3 h | 0.42 | 1.53 (1.21–1.92) | **<0.001** |
|     >3 h | 0.21 | 1.24 (0.62–2.46) | 0.545 |
| Have any of your family, friends, or classmates gotten or reserved the HPV vaccine? | | | |
|     No | Ref | | |
|     Yes | 0.35 | 1.42 (1.15–1.75) | **<0.001** |
| **eHealth Literacy Scores** | 0.03 | 1.03 (1.01–1.05) | **0.002** |

Notes.
Nagelkerke's $R^2 = 0.047$, chi-square $= 4.478$, $df = 7$, $p$ value $= 0.723$.
Bold indicates statistical significance ($p < 0.05$).

moderate daily exercise and peers' HPV vaccination status. Compared with undergraduate students, male graduate students presented significantly greater willingness to receive the HPV vaccine (OR: 1.82; 95% CI [1.00–3.32]; $p = 0.049$). College students with a family history of cancer presented significantly greater willingness to receive HPV vaccination (OR: 1.29; 95% CI [1.01–1.66]; $p = 0.041$). Participants who engaged in moderate daily exercise (1–3 h per day) were more likely to be willing to receive the HPV vaccine (OR: 53; 95% CI [1.21–192]; $p < 0.001$) than those who exercised less (<1 h per day). However, no significant difference was observed in the willingness to receive HPV vaccination between students who exercised at a very high level (>3 h per day) and those with lower levels of physical activity ($p = 0.545$). Additionally, students with close family members who had received or scheduled HPV vaccination demonstrated significantly greater willingness to be vaccinated compared to those without such family influence (OR: 1.42; 95% CI [1.15–1.75]; $p < 0.001$).

Moreover, the participants' HL positively influenced their willingness to receive HPV vaccination (OR: 1.03; 95% CI [1.01–1.05]; $p = 0.002$).

### Van Westendorp PSM

Among 781 participants in the Van Westendorp PSM questionnaire who expressed willingness to self-fund HPV vaccination, vaccine type preferences emerged as follows: 6.8% (53/781) selected imported quadrivalent vaccine, 78.7% (615/781) opted for imported

**Table 5  Van Westendorp price points (in CNY).**

| Price points | Quadrivalent HPV vaccine (imported) | Nine-valent HPV vaccine (imported) |
|---|---|---|
| Highest Pricing Point (HPP) | 1,012.50 | 1,194.44 |
| Acceptable Price Point (APP) | 931.58 | 1,070.26 |
| Optimal Price Point (OPP) | 925.00 | 1,073.61 |
| Lowest Pricing Point (LPP) | 830.36 | 955.00 |
| Market Price | 831 | 1,331 |
| Acceptable Price Range (APR) | [830.36, 1,012.50] | [955.00, 1,194.44] |

nine-valent vaccine, and 14.5% (113/781) chose other HPV vaccine types. Therefore, in the subsequent PSM analysis, the quadrivalent and nine-valent vaccine groups consisted of 53 and 615 male medical college students, respectively.

The APRs (between the lowest and highest price points, in CNY) identified for the quadrivalent and nine-valent HPV vaccines were [830.36, 1,012.50] ([\$116.62, \$142.21]) and [955.00, 1,194.44] ([\$134.13, \$167.76]). Notably, the market price of the nine-valent HPV vaccine was 1,331.00 CNY (\$187.20), which exceeds the APR for the male college student population. However, the market price of the quadrivalent HPV vaccine (831.00 CNY, \$116.71) decreased within the APR and closely aligned with the LPP (equivalent proportions of "too cheap/expensive" perceptions at this price point) while remaining below the OPP (equivalent proportions of "too cheap/too expensive" perceptions at this price point) (Table 5).

To more clearly understand male college students' price sensitivity toward different HPV vaccines, four cumulative response curves were generated from the data. Plots of the data for the Van Westendorp PSM are presented in Fig. 1 for the quadrivalent HPV vaccine (imported) and in Fig. 2 for the nine-valent HPV vaccine (imported).

For the quadrivalent HPV vaccine, the OPP (the intersection of the "too cheap" and "too expensive" curves) was approximately 925.00 CNY (\$129.92), which closely aligned with the APP (the intersection of the "cheap" and "expensive" curves) pricing of 931.58 CNY (\$130.84). Similarly, with respect to the nine-valent HPV vaccine, the OPP was approximately 1,073.61 CNY (\$150.79), which closely aligned with the APP of 1,070.26 CNY (\$150.32) (Figs. 1 and 2). Overall, the Van Westendorp PSM analysis revealed high price sensitivity within the study population, as evidenced by the minimal difference observed between the OPP and APP. These findings may suggest that male college students exhibit significant responsiveness to price adjustments for both quadrivalent and nine-valent HPV vaccines.

## DISCUSSION

### Willingness to receive HPV vaccination

This study represents the first large-sample cross-sectional survey on HPV vaccination conducted among college students at Guangdong Medical University and addresses the gap in related research in colleges and universities in the Dongguan area, China. The findings indicate that 59.4% of male college students are willing to receive the HPV
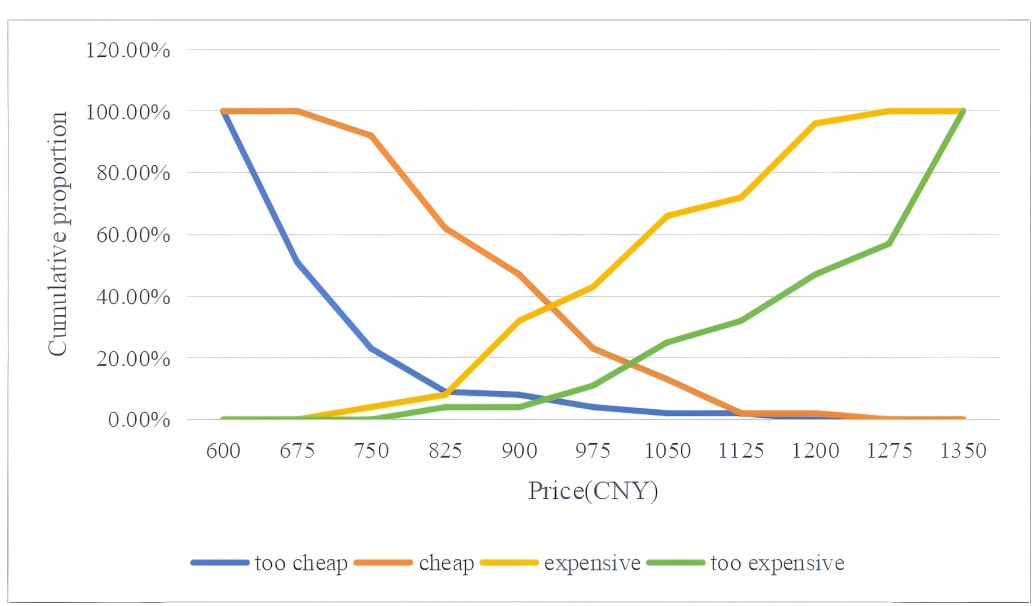

**Figure 1** Van Westendorp price sensitivity graphs for the quadrivalent HPV vaccine (imported).

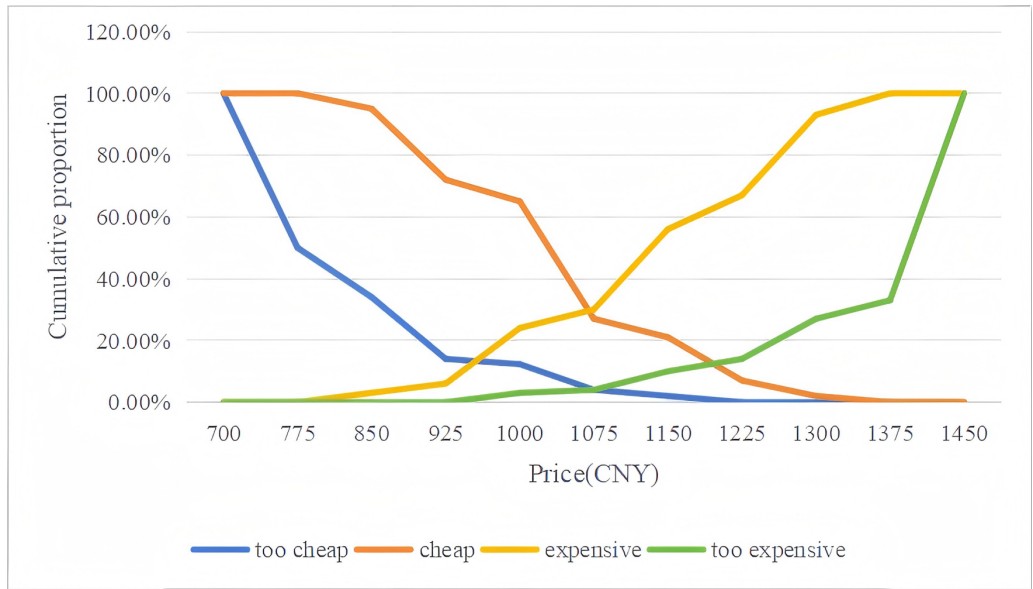

**Figure 2** Van Westendorp price sensitivity graphs for the nine-valent HPV vaccine (imported).

vaccine. However, this percentage is lower than that reported in similar domestic studies, such as studies conducted in Ningxia (80.3%) (*Xiang-rong et al., 2023*) and Zhejiang (68.9%) (*Wang, 2024*). This figure is also lower than findings from international studies in countries such as Italy, South Korea, and the United States, where male college students exhibited high willingness to receive the HPV vaccine with reported rates ranging from

62% to 79% (*Choi & Park, 2016*; *Daley et al., 2010*; *Mascaro et al., 2019*). These results indicate that there are notable disparities in the willingness to receive HPV vaccination across different countries or regions. Therefore, it is necessary to conduct context-specific analyses and develop targeted measures to increase the motivation of male medical college students to receive HPV vaccination.

## Socioeconomic and demographic characteristics

Our study identified several key factors associated with HPV vaccination intentions among male medical college students, including education level, family history of cancer, daily exercise habits, and the vaccination status of individuals within their social circles. These findings provide valuable insights for designing targeted interventions to improve vaccine uptake in this population.

First, graduate students were more willing to receive the HPV vaccine than undergraduates, which is consistent with previous research (*Zhang et al., 2022*). This may be attributed to the advanced education level of graduate students, which likely facilitates greater knowledge acquisition and a deeper understanding of HPV, its associated diseases, and the benefits of vaccination. Educational interventions tailored to undergraduates, such as campus-wide health seminars or digital campaigns, may help to bridge this knowledge gap and increase vaccination intentions. However, caution is warranted when interpreting findings related to postgraduate students, given their limited representation (constituting merely 3.74% of the total cohort). Although statistically significant differences were detected, the restricted subgroup size may potentially exaggerate effect size estimates and compromise the generalizability of these specific results.

Second, students with a family history of cancer showed increased vaccination willingness, which aligns with the findings of previous studies (*Jia et al., 2023*). Personal exposure to cancer within the family may heighten perceived risk and motivate preventive health behaviors, including vaccination (*Brewer & Fazekas, 2007*). Public health messaging can leverage this finding by emphasizing the role of HPV vaccination in reducing the risk of cancer, particularly for individuals with a family history of HPV-related cancers.

Third, we found a positive association between moderate daily exercise and vaccination intentions, suggesting that health-conscious behaviors reinforce one another (*Bohn-Goldbaum et al., 2022*). Regular exercisers often exhibit greater health awareness and a stronger sense of responsibility for their well-being, making them more likely to adopt preventive measures such as vaccination (*Zhang et al., 2024*). Integrating exercise promotion into HPV vaccination campaigns, such as through fitness challenges or partnerships with campus sports programs, can further increase vaccine uptake.

Finally, the vaccination status of individuals within students' social circles significantly influenced the students' own vaccination intentions, consistent with prior findings (*Jia et al., 2023*). Observing peers, family members, or colleagues receiving the vaccine may alleviate safety concerns, boost confidence in vaccination, and increase its social acceptability. Encouraging vaccinated individuals to share their experiences through social media or peer-led initiatives may amplify this effect and foster a culture of vaccination.

In conclusion, our findings highlight the multifaceted nature of HPV vaccination intentions among male college students. Addressing these factors through targeted educational campaigns, risk communication strategies, health-promoting activities, and peer influence interventions can significantly improve vaccine coverage in this key demographic.

## Health literacy

Our study revealed a significant positive association between HL and willingness to receive the HPV vaccine, consistent with previous research (*Bhoopathi, Bhagavatula & Singh, 2022*; *Galvin et al., 2023*). Higher HL levels are linked to improved health-related behaviors and outcomes (*Sun et al., 2025*), including a greater inclination to invest in preventive health measures such as vaccination. Research indicates that higher levels of HL correlate with improved comprehension of HPV-related information, including vaccine efficacy and safety profiles (*Albright & Allen, 2018*). This increased understanding helps to dispel vaccine-related misinformation and concerns and enables individuals to make evidence-based choices about immunization that serve their personal health objectives (*Malik et al., 2024*). Therefore, future efforts should prioritize targeted educational campaigns designed to improve HL. For example, campus-based workshops, eHL modules, and peer-led educational programs can be implemented to address specific knowledge gaps and misconceptions about HPV and its vaccination. Additionally, collaboration with healthcare providers to deliver clear, culturally sensitive, and age-appropriate information can further empower students to make informed decisions. By fostering a health-literate population, these interventions have the potential to significantly increase HPV vaccination rates among male college students.

## Van Westendorp PSM

The Van Westendorp PSM analysis revealed that male college students presented high price sensitivity toward both the quadrivalent and nine-valent HPV vaccines. Comparative surveys have demonstrated that domestic vaccines have lower acceptable price ranges (CNY 910.63–2,866.96) than their imported counterparts (CNY 1,689.80–3,252.43) (*Zhou et al., 2022*), providing quantitative evidence for price differentials between vaccine types. While the high cost of HPV vaccines remains a significant barrier to accessibility in China (*You et al., 2024*), our findings suggest that the price gap between the acceptable and market prices of HPV vaccines has gradually narrowed, particularly for the quadrivalent vaccine (*Lu et al., 2022*; *Wang et al., 2022*). Specifically, the market price of the quadrivalent HPV vaccine (CNY 831.00, $116.71) generally decreased within the APR (CNY 830.36–1,012.50, $116.62–142.21) in our study and approached the LPP (CNY 830.36, $116.62), indicating that price may no longer be the primary barrier for this type of vaccine. This convergence is further supported by a 2021 national survey of healthcare workers reporting median willingness-to-pay values (CNY 1,250–1,400) that were 18–30% below the then-prevailing market prices (*Lu et al., 2022*). In contrast, the nine-valent vaccine remains priced (CNY 1,331.00, $187.20) above students' APR (CNY 955.00–1,194.44, $134.13–$167.76) in our study, with the current market price exceeding the upper bound of reported acceptable

ranges by 11.4–39.4%, highlighting the need for targeted pricing strategies or subsidies to improve the affordability of the vaccine.

The Van Westendorp PSM analysis revealed distinct price sensitivity patterns for quadrivalent and nine-valent HPV vaccines among male college students. For quadrivalent vaccines, interventions should focus on educational campaigns to address nonprice factors such as safety perceptions and HL. For the nine-valent vaccine, cost-reduction strategies such as subsidies or tiered pricing are essential in addition to efforts to improve health literacy and awareness, to increase vaccine uptake.

### Limitations of the study

This study also has limitations. First, the convenience sampling approach and the overrepresentation of medical students (75% of participants) may introduce sampling bias, potentially inflating health literacy estimates compared to the general male student population. Consequently, findings primarily reflect trends among medical students and require validation in more diverse academic cohorts. Second, the reliance on self-reported data (*e.g.*, vaccination intentions and health literacy assessments) may be subject to recall bias and social desirability bias, particularly given the sensitive nature of health-related decision-making. Third, the generalizability of the findings may be limited by the cross-sectional design and the regional specificity of the sample. Fourth, while we controlled for key covariates including age and education level, other potential confounders (*e.g.*, socioeconomic status, cultural attitudes toward vaccination) were not systematically measured, which may influence the interpretation of price sensitivity patterns. Fifth, the sole reliance on HL limits theoretical grounding by excluding broader models (*e.g.*, Health Belief Model or Theory of Planned Behavior) addressing psychosocial influences. Sixth, since HPV vaccination for males (both adults and boys) was not yet available in China at the time of this survey, the responses from male college students should be interpreted as simulated rather than actual behavioral data. Finally, the investigation was limited to assessing one-time purchase intentions. The relatively limited sample size for recipients of the quadrivalent HPV vaccine compared with the size of the nine-valent vaccine group may affect the robustness of our findings. These limitations underscore the need for subsequent studies with larger, more balanced sample sizes to validate the applicability of the Van Westendorp PSM in vaccine-related consumer research (*Lipovetsky, Magnan & Zanetti-Polzi, 2011*). With the approval of the quadrivalent HPV vaccine for males aged 9–26 years in mainland China on January 8, 2025, this study offers preliminary validation of the Van Westendorp PSM methodology and introduces a novel approach to research on HPV vaccine pricing.

## CONCLUSIONS

This study assessed the willingness to receive HPV vaccination and price sensitivity among Chinese male medical college students in Dongguan. The findings revealed that more than half of the participants were willing to receive the HPV vaccine. This willingness was influenced by education level, family history of cancer, daily exercise, peers' vaccination status, and HL. While the price of the quadrivalent vaccine aligned with students' acceptable

range, the nine-valent vaccine remained costly. To improve uptake, interventions should combine price reduction strategies for the nine-valent vaccine with educational campaigns to increase HL and address nonprice barriers, particularly for the quadrivalent vaccine.

## ACKNOWLEDGEMENTS

We sincerely appreciate the invaluable support and contributions of numerous teachers and students at Guangdong Medical University. We are grateful to all those who assisted with data collection, provided survey instruments, and participated in the survey.

### Funding
This work was supported by JST SPRING (No. JPMJSP2132) and the Dongguan Sci-tech Commissioner Program (No. 20231800500372). The funders had no role in study design, data collection and analysis, decision to publish, or preparation of the manuscript.

### Grant Disclosures
The following grant information was disclosed by the authors:
JST SPRING: No. JPMJSP2132.
the Dongguan Sci-tech Commissioner Program: No. 20231800500372.

### Competing Interests
The authors declare there are no competing interests.

### Author Contributions
- Yuan Li conceived and designed the experiments, performed the experiments, analyzed the data, prepared figures and/or tables, authored or reviewed drafts of the article, and approved the final draft.
- Hiromi Kawasaki conceived and designed the experiments, authored or reviewed drafts of the article, and approved the final draft.
- Zhengai Cui conceived and designed the experiments, analyzed the data, authored or reviewed drafts of the article, and approved the final draft.
- Sae Nakaoka analyzed the data, authored or reviewed drafts of the article, and approved the final draft.

### Human Ethics
The following information was supplied relating to ethical approvals (i.e., approving body and any reference numbers):

This study was approved by the Clinical Research Ethics Committee of the Affiliated Hospital of Guangdong Medical University (ethical application ref: KT2023-126-01, approval number: PJKT2023-126).

## Ethics

The following information was supplied relating to ethical approvals (i.e., approving body and any reference numbers):

This study was approved by the Clinical Research Ethics Committee of the Affiliated Hospital of Guangdong Medical University (ethical application ref: KT2023-126-01, approval number: PJKT2023-126).

## Data Availability

The raw measurements are available in the Supplementary File.

## Supplemental Information

Supplemental information for this article can be found online at http://dx.doi.org/10.7717/peerj.19699#supplemental-information.

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
