# Peer review of "Factors associated with the intention to vaccinate and price sensitivity to the human papillomavirus (HPV) vaccine among Chinese male medical college students: a cross-sectional survey"

_PeerJ, doi:10.7717/peerj.19699_

## Round 0.1 · original submission · Minor Revisions

The study is generally well conducted, and the writing style is clear and unambiguous.

However, in addition to the remarks of the reviewers, I found some gaps that should be addressed:

1. In the abstract, you should:
- Add the tool used to distribute the questionnaire and the study period in the methods.
- Add the total number of participants and the quantitative values (OR) in the results.
- Revise the conclusion by providing the most important results.

2. The introduction needs some amendments and organization of the sequencing of the ideas, and try to better emphasize the hypothesis of the work. You should also avoid the use of non-referenced sentences. (Lines 62, 65...). Also, in line 65 (few studies...you should cite them).

3. In the methods, you should highlight the subtitles (lines 102, 107, 114).

4. In the results, you should write the "p" in lowercase.

5. For the discussion, you should provide more quantitative results. The comparisons are vague and should be completed with the quantitative results.

6. At last, the manuscript should be edited for multiple grammatical errors.

**Language Note:** The Academic Editor has identified that the English language must be improved. PeerJ can provide language editing services - please contact us at [email protected] for pricing (be sure to provide your manuscript number and title). Alternatively, you should make your own arrangements to improve the language quality and provide details in your response letter. – PeerJ Staff

·

Basic reporting

Generally the writing style is clear and unambiguous. Some editing and proof reading will be required.

Experimental design

The design seems appropriate. However, I have the following questions:
1. Did the study participants have a complete idea of the HPV vaccine, HPV related diseases, their treatment etc.before they expressed the willingness to get vaccinated or to buy?
2. What precaution was taken to ensure that the price point expressed was realistic and the participants had consideration for other opportunity costs based on their current earnings and expenses.
3. What mechanisms were taken to avoid biases like "yea saying" or protest responses

Validity of the findings

1. The sample had less than 4% postgraduate students. The validity of the finding that Post Grads had a higher price point or willingness to get vaccinated may be exaggerated.
2. 75% of the sample population were medical majors. This is definitely not a representative sample of Chinese men. This would severely influence the major variable of Health Literacy.

Additional comments

The authors can look at changing the title of the manuscript and target it to Male Medical Majors rather than Chinese men.

Reviewer 2 ·

Basic reporting

The manuscript is generally well-written, with professional and clear English.
The introduction provides relevant background but does not clearly articulate a defined knowledge gap. For example, the authors state that "few domestic studies have explored the association between HL and HPV vaccination intentions," but they do not clearly explain what those gaps are or how this study advances knowledge beyond prior work.
The structure of the paper aligns with standard scientific reporting, but some reporting elements are missing. Counts are sometimes reported inconsistently as percentages (N%) without corresponding absolute values (n/N) in tables or results.
Ethical approval and informed consent procedures are adequately described and documented.
The STROBE checklist is included but does not address missing data or provide a participant flow diagram, which limits transparency.

Experimental design

The study is described as using "random sampling," but the method is not clearly detailed. In reality, participants were recruited via WeChat using an online platform, which aligns more with convenience sampling, not true random sampling.
The exclusion criteria are unusual: removing respondents based on short survey completion time and inconsistent answers is valid for quality control, but more transparency is needed on how many were excluded under each criterion.
The questionnaire appears to have been appropriately adapted, and permission for instrument use (eHealth literacy scale) is provided.
The use of the van Westendorp Price Sensitivity Meter is a novel and appropriate approach for assessing vaccine affordability perception.

Validity of the findings

The analysis is statistically sound, but some issues impact the overall validity.
Missing data management is not described. There is no indication of whether there were any missing values, how many participants were excluded at each stage, or whether imputation or sensitivity analyses were performed.
The discussion of limitations is appropriate, but it fails to acknowledge the sampling bias, self-reported nature of data, or lack of control for confounders beyond HL.
The study uses only HL as a theoretical lens, and does not employ any broader health behavior models such as the Health Belief Model or Theory of Planned Behavior, which limits interpretability and theoretical grounding.

---

## Round 0.2 · Minor Revisions

The introduction still needs some amendments. Please find my comments in the attached file

·

Basic reporting

Appropriate

Experimental design

Appropriate. Comments well responded to

Validity of the findings

Comments well responded to

Additional comments

None

---

## Round 0.3 · accepted · Accept

The authors have addressed all comments.